# Understanding the representativeness of FLUXNET for upscaling carbon flux from eddy covariance measurements

Jitendra Kumar<sup>1</sup>, Forrest M. Hoffman<sup>2</sup>, William W. Hargrove<sup>3</sup>, and Nathan Collier<sup>2</sup> <sup>1</sup>Environmental Sciences Division, Oak Ridge National Laboratory, Oak Ridge, TN, USA

<sup>2</sup>Computer Science and Mathematics Division, Oak Ridge National Laboratory, Oak Ridge, TN, USA

<sup>3</sup>Eastern Forest Environmental Threat Assessment Center, USDA Forest Service, Asheville, NC, USA

Correspondence to: Jitendra Kumar (jkumar@climatemodeling.org)

Abstract. Eddy covariance data from regional flux networks are direct *in situ* measurement of carbon, water, and energy fluxes and are of vital importance for understanding the spatio-temporal dynamics of the the global carbon cycle. FLUXNET links regional networks of eddy covariance sites across the globe to quantify the spatial and temporal variability of fluxes at regional to global scales and to detect emergent ecosystem properties. This study presents an assessment of the representativeness of

- FLUXNET based on the recently released FLUXNET2015 data set. We present a detailed high resolution analysis of the evolving representativeness of FLUXNET through time. Results provide quantitative insights into the extent that various biomes are sampled by the network of networks, the role of the spatial distribution of the sites on the network scale representativeness at any given time, and how that representativeness has changed through time due to changing operational status and data availability at sites in the network. To realize the full potential of FLUXNET observations for understanding emergent ecosystem
- properties at regional and global scales, we present an approach for upscaling eddy covariance measurements. Informed by the representativeness of observations at the flux sites in the network, the upscaled data reflects the spatio-temporal dynamics of the carbon cycle captured by the *in situ* measurements. This study presents a method for optimal use of the rich point measurements from FLUXNET to derive an understanding of upscaled carbon fluxes, which can be routinely updated as new data become available, and direct network expansion by identifying regions poorly sampled by the current network.

Data from this study are available at http://dx.doi.org/10.15486/NGT/1279968

# 1 Introduction

Terrestrial ecosystems are a critical component of the global carbon cycle regulating the climate through a range of biogeophysical and biogeochemical mechanisms. Terrestrial ecosystems exchange about 120 Pg of carbon per year with the atmosphere,

through the processes of photosynthesis and respiration (Schlesinger and Bernhardt, 2013). Eddy covariance-based methods are widely used around the world to measure the gas exchange between vegetation and the atmosphere. The eddy covariance method assumes the measurement site is located in flat terrain, experiences steady or stable atmospheric conditions, and is surrounded by uniform vegetation for an extended distance in the upwind direction (Baldocchi, 2003). In practice, however, sites are often located in non-ideal and spatial heterogeneous regions and are thus may be prone to varying degrees of measurement

# Science Scienc

error (Baldocchi, 2003). Understanding the spatial representativeness of site measurements is required for landscape-scale integration and interpretation of carbon flux measurements. While an individual site may be considered representative of the biome in which it is located, any carbon flux measurement network must include a broad representation of vegetation types, climates, and stages of disturbance. FLUXNET is a global network of micrometeorological flux measurement networks consisting of in-

- dividual sites that measure the exchange of carbon dioxide, water vapor, and energy between the biosphere and the atmosphere (Baldocchi et al., 2001). FLUXNET consists of more than ~750 sites (http://fluxnet.fluxdata.org/sites/historical-site-status/) across the globe that are independently operated by regional networks (http://fluxnet.fluxdata.org/about/regional-networks/). However, the locations of the sites in the network were not formally designed to uniformly and consistently observe global biomes and thus represent a sparse and spatially biased sampling of the global terrestrial ecosystem. Careful synthesis of spatio-
- temporally sparse flux observations is required to understand the regional to global scale carbon cycle. Upscaling of fluxes to regional to global scales requires understanding and quantification of the spatial representativeness of the global network of flux sites.

A number of studies have analyzed the representativeness of regional networks of flux sites. Hargrove et al. (2003) analyzed the representativeness of the AmeriFlux network using an ecoregion-based approach and found that while most of the conti-

- nental United States was well represented by the network, the Pacific Northwest and Texas grasslands were poorly represented by the network sites. Yang et al. (2008) conducted a similar independent study of AmeriFlux network repsentativeness using remotely sensed climate and vegetation data products and arrived at conclusions similar to Hargrove et al. (2003). Hargrove and Hoffman (2005) presented a quantitative ecoregion-based approach to assess the representiveness of networks and a method for design of sampling networks. Sulkava et al. (2011) studied the representativeness of the European flux tower network and
- developed a quantitative network design tool for identifying and prioritizing addition of new flux sites. Chen et al. (2011, 2012) used remote sensing and footprint analysis to characterize the spatial representativeness of flux sites across the Canadian Carbon Program Network. He et al. (2015) conducted a similar assessment of regional representativeness for the Chinese flux network. They also identified a set of sites that would complement the existing flux network. Most of these assessments of the spatial representativeness of regional flux networks across the globe have been focused on quantifying the coverage by the net-

work and for the design of new sampling sites. However, the quantitative representativeness of network sites has not yet been applied to synthesize and upscale observations beyond the footprints of individual sites to provide landscape- to global-scale estimates of flux measurements.

We present here an analysis of the quantitative representativeness of the FLUXNET global network of sites that contributed to the FLUXNET2015 data set. While a large number of global flux sites are affiliated with the FLUXNET network, only a

- subset of site data are typically available in data collections. Thus, only the sites with data available in the most recent July 12, 2016, release of FLUXNET2015 were included in our study. Independently operated sites in the FLUXNET network often vary significantly in their period of active operation or data availability for logistical reasons. Thus in this study, in addition to quantifying the spatial representativeness of the network, we quantify the temporally variable representativeness of data availability within the FLUXNET2015 collection. Finally, we present an approach for global upscaling of eddy covariance
- fluxes from FLUXNET network informed by the temporally varying spatial representativeness of the flux sites.

(a) This map shows the spatial locations of all FLUXNET sites. Red triangles indicate the location of (b) Active FLUXNET2015 sites each year during the all sites registered in the FLUXNET network. Blue circles indicate sites included in the FLUXNET2015 data set. The size of the blue circles quantifies the period of operation or data availability.

Figure 1. Eddy covariance sites in the FLUXNET2015 collection.

# 2 Data and Methods

# 2.1 Eddy covariance measurements

FLUXNET provides uniform and high quality data products through coordination among various regional flux networks across the globe (http://fluxnet.fluxdata.org/data/fluxnet2015-dataset/data-processing/). We used the latest release of the FLUXNET2015

5 eddy covariance data set collection in this study. The time series products in the FLUXNET2015 collection were gap filled (Reichstein et al., 2005) and went through a rigorous QA/QC procedure (Pastorello et al., 2014). The model efficiency method reference GPP product (GPP\_DT\_CUT\_REF) calculated using day time data (Lasslop et al., 2010) was used to create an upscaled data product in this study.

While more than ~750 sites are registered in the FLUXNET database (http://fluxnet.fluxdata.org/sites/historical-site-status/),
indicated by red triangles in Figure 1(a), data from only 165 global sites were available in the FLUXNET2015 data set released July 12, 2016, and shown by filled blue circles in Figure 1(b). This study was limited to data contained in the FLUXNET2015 data set. Environmental data sets used in our study were not available at the permanent wetland site at Adventdalen, Norway (NO-Adv) and thus was excluded from the analysis. Thus, the GPP observations from 164 sites globally (Figure 1(a)) spanning the period 1991–2014 were used. FLUXNET sites are maintained and operated by independent groups and the period of active

15 operation and data availability varies widely from site to site, ranging from a single year to up to 24 years (Figure 1(a)). Figure 1(b) shows the number of active sites for which data was available in the FLUXNET2015 collection in any year during the period 1991–2014. Due to extremely limited data availability before year 2000, we limited our study to the period 2000–2014. There also exists some variability in variables and data fields available from any site. Figure 1(b) also shows the number of sites where the GPP\_DT\_CUT\_REF variable used in this study was available.

# 2.2 Environmental variables

Ecosystem structure and functioning are governed by multiple independent control variables including climate, parent material, topography, potential biota and time (Chapin et al., 2012). A set of environmental variables to capture climatic, edaphic and topographic characteristics were selected to characterize the terrestrial ecosystem in this study (Table 1).

Table 1. Environmental variables used for ecoregion delineation, representativeness analysis and upscaling. These data are in the form of  $\sim$ 4 km raster grids.

| Variable Description                          | Units             | Source                                                  |
|-----------------------------------------------|-------------------|---------------------------------------------------------|
| Bioclimatic Variables                         |                   |                                                         |
| Annual mean temperature                       | °C                | Hijmans et al. (2005)                                   |
| Mean diurnal range                            | °C                | Hijmans et al. (2005)                                   |
| Isothermality                                 | -                 | Hijmans et al. (2005)                                   |
| Temperature seasonality                       | °C                | Hijmans et al. (2005)                                   |
| Temperature annual range                      | °C                | Hijmans et al. (2005)                                   |
| Mean temperature of wettest quarter           | °C                | Hijmans et al. (2005)                                   |
| Mean temperature of driest quarter            | °C                | Hijmans et al. (2005)                                   |
| Mean temperature of warmest quarter           | °C                | Hijmans et al. (2005)                                   |
| Mean temperature of coldest quarter           | °C                | Hijmans et al. (2005)                                   |
| Annual precipitation                          | mm                | Hijmans et al. (2005)                                   |
| Precipitation during the wettest quarter      | mm                | Hijmans et al. (2005)                                   |
| Precipitation during the driest quarter       | mm                | Hijmans et al. (2005)                                   |
| Precipitation during the warmest quarter      | mm                | Hijmans et al. (2005)                                   |
| Precipitation during the coldest quarter      | mm                | Hijmans et al. (2005)                                   |
| Edaphic Variables                             |                   |                                                         |
| Available water holding capacity of soil      | mm                | Global Soil Data Task Group (2000); Saxon et al. (2005) |
| Bulk density of soil                          | g/cm <sup>3</sup> | Global Soil Data Task Group (2000); Saxon et al. (2005) |
| Soil carbon density                           | $g/m^2$           | Global Soil Data Task Group (2000); Saxon et al. (2005) |
| Total nitrogen density                        | $g/m^2$           | Global Soil Data Task Group (2000); Saxon et al. (2005) |
| Topographic Variables                         |                   |                                                         |
| Compound topographic index (relative wetness) | -                 | Saxon et al. (2005)                                     |

<sup>5</sup> Hijmans et al. (2005) developed very high resolution interpolated climate surfaces (WorldClim) for global land areas. Using a large number of global sources of weather station data, they created global climate surfaces at 30 arc second spatial resolution ( $\sim$ 1 km). They also derived a set of 19 bioclimatic variables (BioClim) (with spatial resolution of 2 arc minute or  $\sim$ 4 km) that are biologically meaningful, representing annual trends, seasonality and extreme or limiting environmental factors. We selected 14 bioclimatic variables (Table 1) to represent the global bioclimatic environment in this study. We also selected four

<sup>10</sup> edaphic variables (Global Soil Data Task Group, 2000; Saxon et al., 2005) and one topographic variable (Saxon et al., 2005) to characterize not just the above-ground but also the below-ground environment.