# Peer review of "Understanding the representativeness of FLUXNET for upscaling carbon flux from eddy covariance measurements"

_Earth System Science Data, 2016_

## Referee Comment (RC1) · Anonymous Referee #1 · 31 Aug 2016

This study presents an assessment of the representativeness of FLUXNET based on the recently released FLUXNET2015 data set. In light of their past work looking at the representativeness of the AmeriFlux (Hargrove et al., 2003) and 2007 Fluxnet (Sundareshwar et al., 2007)networks, this is a timely and important effort. Studies like this inform users of such data about how representative their data may be beyond the narrow confines of a flux footprint. And there is precedence indicating that similar forests may function similarly despite wide differences in distance, as is case of the comparison of carbon fluxes of beech forests in France and Denmark (Granier et al., 2002). Such a study is also critical for the use and interpretation of flux data by global modeling exercises. We know we can't measure fluxes everywhere, but if the sparse network

is representative of greater climate and ecological spaces, then these measurements can infer information about locales where no measurements exist.

Introduction

The authors do a good job justifying the topic of the paper and describing work that has been done. This effort is especially timely since the FLUXNET 2015 dataset was released this past July. As more and more scientists use this data, having a paper discussing the representativeness of these data will be very useful to the synthesis, remote sensing and modeling communities.

Materials and Methods

The method relies on maps of climate, soils and ecosystems, and given this information it tells us how well the current networks are sampling the ecosystems of the globe. Such an effort is instructive for guiding the placement of additional towers and indicating a priority.

Of course the method is not perfect and is lacking in assessing the representativeness of sites given similar stand age and time since disturbance. Incorporating stand age maps is a direction to think about in future incarnations of this work (Pan et al., 2011). The approach also misses out on the complex mapping of crops in the corn/soybean belt. Since C3 and C4 crops differ so much in their photosynthetic potential and net and gross carbon fluxes (Suyker and Verma, 2012), one could expect sizable uncertainties associated with maps in this region. I'd like to see the authors give an open and unbiased critique of the pros and cons of their representative mapping method and where the approach needs to go into the future as we glean more and better information of the Earth system.

A strength of this work is its discrimination and analysis of the potential (with 700+) network and the actual (160+) network in the living database. Efforts like this can be important for recruiting more outstanding data into the global fold of information.

[Figure]

The paper does a nice job producing maps of these different networks.

Much of this analysis rests on a diverse list of bioclimatic variables, mapped at 4 km resolution. It is unclear if all variables are needed, what their uncertainty is as they are spatially filled, and if all the variables are needed. In addition, in some regions seasonal climate may be better than annual averages or sum. Here, I applaud the authors as they consider quarterly statistics and are agnostic to the timing, giving them flexibility for north and south hemispheres and Mediterranean ecosystems with a winter/spring growing season.

Splitting and lumping is always an intense argument among ecologists. Parsimony has its strength and here I see 10 ecoregions are describes. More information on why this value would be helpful. In other words, what happens if there are more or less?

Figure 3. I like the examples of how representative 2 candidate sites may be. However the authors need to do a better job instructing the reader on the nuance of the grey scale. Is 0 good, why, and how bad is 0.10? In other words we need some better context to interpret the numerical values of the grey scale. A rainbow color map may be more instructive. Plus the figure is vague to what the grey scale is. Nothing is said in the figure caption, nor does the text clearly state if it relates to any equation in particular. While I like the topic, the authors need to work a bit more on being clear and telling a clearer narrative.

Figure 4 is very important as it tells us how representative the 164 site dataset is. Now in reality there remain data in the 2007 La Thuile dataset that for practical reasons did not migrate to the 2015 Fluxnet database, but are still publicly available. So seeing a map with sites from both datasets could be instructive, too. Please add an additional map with the sites in the 2007 and 2015 datasets. The range of the grey scale in Figure 4 is only between 0 and 0.025. What does this mean? Again a rainbow color palette may be better in distinguishing representativeness better. At present all I can assume is white is good.
I would also like to see a potential representative map based on all the registered sites and compare them with the actual network. This is missing and needed information by this analysis.

It is interesting to see the missing regions like Latin America, parts of Asia, Africa and the Pacific Northwest. Now FLUXNET Canada had a number of sites in the Pacific Northwest. I would hope this dark region would lighten up with their inclusion.

Figure 5 is interesting. I like the more detailed and in depth analysis of this paper and looking at the history and growth of the network.

Figures 1 and 6 and Table 2 tell us nothing about the assigned ecoregions in the maps. The authors need to pay more attention to specific details. We need either a table, legend or caption telling us what regions 1 through 10 refer to. More attention to detail is needed here and throughout the paper.

Table 2 is interesting and important in terms of documenting ecoregions that are relatively over and under sampled. Again this analysis is important for helping the network evolve and improve, either with new sites or better recruiting of existing data from under reported regions. We do know there are many more flux towers across China and Brazil and a few in Argentina and Equador. So there is hope.

This paper does something new and different from the earlier incarnations by assessing the representativeness of GPP. On one hand, this maybe is a good idea, given the complex use of climate, flux networks and remote sensing by Jung et al, Beer et al and Xiao et al to produce both global and continental flux maps. Yet, as I read the material I see a lacking of connection between the first part of the paper and this part. The question to me revolves around the uncertainty in global GPP given the actual flux network. I think this team would do better service comparing estimates of GPP with a sparse actual network and with the ideal denser registered network. As I read on and came back to this section, I think my problem in understanding what is going on stems from the very, very vague description of what this team did to 'upscale GPP', compared

to the MTE approach of Jung et al. Much more and better information is needed for understanding what you did.

Results Figures 7 and 8, the legend is impossible to read as the numbers are too faint. Please consider reproduction. Again I feel that the authors have gone off on a tangent and I would argue that they redefine and confine the focus towards representativeness and how increasing and decreasing sites affect integrations of GPP. The comparison here with the MTE approach should be removed and can be the topic of a different independent paper. Careful editing is needed to tell a consistent story. I suggest referring to Josh Schimel's excellent book on writing science.

Figure 9 would be more informative if you could compare information for these latitude belts given the number of towers and uncertainties associated with more or less towers. As you know the tropics are sparse with towers so their uncertainty is high relative to what one may deduce from MTE. Try and do this type of representative analysis. Figures 10 and 11 seem to be a stab at doing this but maybe rethink how you displace and compare the data.

In conclusion, this paper has many strengths and has potential to be a fine and novel contribution. But it needs editing, revision and refining its analysis. I felt that the second part on GPP went off on a tangent, especially since this section was not introduced or explained well in terms of the new method and why. I think it can be fixed and so recommend major revision. Plus this portion was not well described in terms of operation or why the authors digressed in this direction. it is actually a different paper, and given that this was supposed to be a brief contribution, is very much off target.

Personally if this is to be a brief contribution I think there is enough material to produce representative maps based on the 2015 data set, the combined 2007 and 2015 dataset and all registered sites and follow up with a critique on the strengths and weaknesses of this mapping method and where it can and should go.

Finally most figures need better description, legends and bolder labeling

Specific comments

Line 19, You should find a primary the citation to estimates for global photosynthesis rather than the textbook of Schlesinger and Berkhardt. Many fine papers and a review (Anav et al., 2015) exists on this topic.

References

Anav, A. et al., 2015. Spatiotemporal patterns of terrestrial gross primary production: A review. Reviews of Geophysics: n/a-n/a. Granier, A., Pilegaard, K. and Jensen, N.O., 2002. Similar net ecosystem exchange of beech stands located in France and Denmark. Agricultural and Forest Meteorology, 114(1-2): 75-82. Hargrove, W.W., Hoffman, F.M. and Law , B., 2003. New analysis reveals representativeness of the AmeriFlux network. EOS, 84: 529. Pan, Y. et al., 2011. Age structure and disturbance legacy of North American forests. Biogeosciences, 8(3): 715-732. Sundareshwar, P.V. et al., 2007. ENVIRONMENT: Environmental Monitoring Network for India Science, 316(5822): 204-205. Suyker, A.E. and Verma, S.B., 2012. Gross primary production and ecosystem respiration of irrigated and rainfed maize–soybean cropping systems over 8 years. Agricultural and Forest Meteorology, 165: 12-24.

---

## Short Comment (SC1) · 13 Sep 2016

Dear Authors

I'm one of the people involved in the FLUXNET2015 dataset preparation and for this reason I'm happy to see that the data are used. My comment is not related to the paper scientific content but to an important data policy requirement. As reported in the FLUXNET2015 data policy (http://fluxnet.fluxdata.org/data/data-policy/) when the data are used it is requested "that every publication specify each site used with the FLUXNET-ID, data-years used" and alos when available a reference describing the site. This is the way a PI can track their data use and for this reason important to follow. I would appreciate if you could add this table in the manuscript.

Dario Papale

---

## Referee Comment (RC2) · Anonymous Referee #2 · 28 Sep 2016

This study targets one of the fundamental but still challenging aspects of FLUXNET data sets – the representativeness issue. The authors adopt a similar approach to their previous works, where the k-mean clustering technique is applied to determine the dissimilarity among pixels in the multidimensional environmental space and to assess the representativeness of FLUXNET site locations and available data sets. Based on the data availability of the FLUXNET2015 data set, the study also shows the potentials and limitations of such data set in extrapolating in space and in time.

Overall, I think the manuscript is generally well-crafted, and there are a number of potentially important messages could be delivered and learned by data contributors, managers, and users. For the three proposed objectives of the study (Line 28-35,

Page 2), I think the study is sufficient to answer the first two about the quantification of FLUXNET representativeness and its temporally variable nature. I have some concerns and questions about the third objective, which aims to upscale the data to a much broader extent in space and in time (see comments below). There are a few places in the current manuscript that needs carful revision and re-organization for clarification and avoiding misinterpretation. Here, I list my general and a few specific comments.

General Comments:

[1] My major concern is the justification of the upscaling approach and/or the interpretation of the upscaled products. The authors repeat a couple of times in the manuscript about the potential bias/uncertainty of the upscaling practices. I think the message is important. Given that the original data availability is unevenly distributed in space and in time and the upscaling approach is pure empirical, one shouldn't be too surprised to see the unexpected predictions in areas or periods with limited data (e.g., Figures 7-9). The question is "how would the authors like readers to view and interpret their GPP predictions?" On the one hand, since this is a data-publishing journal, I'd expect a large portion of discussion meant to support or justify the dataset – a new gridded monthly GPP product. If the authors intend to do so, then I think there are several places in the upscaling processes needed to be clearly defined, modified, or better justified. On the other hand, if the goal of the upscaling practices is to demonstrate and address the representativeness of FLUXNET data sets, then the final data products of this study might be more appropriately interpreted as representativeness metrics. Either way would be valuable, but it's problematic trying to do both or something in-between. I'd hold off some specific comments following this thread of discussion for now, as I feel that the authors first need to decide the ultimate goal of the study and data products, and tailor the analysis and manuscript organization accordingly.

Some examples related to the justification of upscaling: How to adequately incorporate phenology in the prediction? How to deal with the difference of land cover (or vegetation cover) within the ecoregions? Certain ecoregions and periods have very sparse data

coverage. Would it be appropriate to upscale in these cases? It may also need a section to discuss the robustness and uncertainty of the upscaling approach, e.g., cross-validation.

[2] The comparison of the upscaled GPP to MTE GPP is interesting, but requires some more works to help clarify the difference. Currently, it's difficult to interpret the difference. Both upscaling products started from different versions of FLUXNET data sets. The algorithms and variables (environmental factors) used in upscaling practices were also different (e.g., different variables, resolutions, and different ways to incorporate the seasonal/interannual dynamics...). A more careful and comprehensive comparison would be valuable, which could help better interpret the data products, to show potential areas needing more data, or to pinpoint the limitations of each upscaling approach.

[3] As also pointed out by others in the short comments, please be sure to follow the data policy of using FLUXNET data sets. There's no site list attached, but it seems the sites used in the study contain both Tier 1 and Tier 2 sites (??), which have slightly different requirements for credits and acknowledgements. Please make sure fulfilling the data policy when revising the manuscript.

Specific Comments

[1] Line 25-27 Page 2: Theoretically, there are at least a few steps involved in upscaling the tower data to a landscape- or global-scale estimate. A number of studies focused on the upscaling from tower footprints (varying through time) to grid pixels that were seen by satellite remote sensing or reanalysis data products. The focus of current study is more on the later steps, which assumes that flux data adequately represent the pixel properties, and exercise the upscaling from pixels to pixels. Please revise the description (here and all followings).

[2] Line 7 Page 3: It depends on the ultimate goals of the study (referred to general comment [1]), but the authors might need to explain shortly or justify why choose

[Figure]

GPP_DT_CUT_REF over other available GPP products in FLUXNET data set.

[3] Table 1: It's not clear about the temporal resolution of these variables. Are they yearly variables (1 value for each pixel for each year), or long-term averages (1 value for each pixel for all years)? These details also miss later in the upscaling practices (e.g., 2.5.1). It's not clear to me how the upscaling is done on monthly time series, as most of the variables listed here are yearly or long-term averages.

[4] Section 2.3, Figure 2: I wonder if it's feasible to introduce IGBP (or other vegetation/land cover gridded products) into the classification or clustering analysis. There are certainly finer-resolution details missing in the current classification, e.g., the Midwest croplands vs Northeast US temperate forests. I wonder whether adding IGBP may or may not help improve the upscaling predictions.

[5] Section 2.4: Some of the descriptions in this section are duplicated to those in the Introduction. Please consider trimming or reorganizing them.

[6] Line 1-20 Page 6: Need a short sentence to describe how the Euclidean distance is translated to dissimilarity or representativeness.
* * *

---

## Referee Comment (RC3) · Anonymous Referee #3 · 7 Oct 2016

Kumar et al present a method and associated datasets for the representativeness of FLUXNET sites as well as a method to produce gridded flux maps based on the representativeness analysis. The topic itself is very important, and the paper is well written. However, I see many caveats of the analysis.

Major points:

Representativeness analysis: Concept: There is no introduction or discussion of what is meant by representativeness. The representativeness of "sites" described by static biotic (e.g. vegetation type) and abiotic (climate, soil, topography) conditions should be related to a certain variable of the sites, e.g. GPP. I would argue that the representativeness of sites with respect to GPP is different from the representativeness of sites

with respect to NEE or with respect to biodiversity or whatever. Suppose GPP would only depend on radiation and suppose NEE would only depend on time since last disturbance, then for GPP representativeness should refer to sampling existing radiation conditions, and for NEE should refer to sampling existing conditions of time since last two disturbance. The two maps would look entirely different. Basically, variables considered in representativeness analysis should be specific to a certain property or flux of interest and only relevant and driving variables should be considered. Here, the authors do the example for GPP and use only static climate, soil, and topographic variables. The selection of variables seems to be ad hoc and whatever is available at high resolution. There are no variables with respect to vegetation properties which would actually be available too like maps of vegetation type, fapar, tree cover fraction etc. Now comes the temporal aspect: one could argue that a site in the temperate zone samples temperature and radiation conditions from "tundra" to "tropical" over the course of the seasons. Which role does this play for representativeness?

Method: The results of the representativeness analysis depend on the variables that are plugged in, their correlation structure, and the chosen distance metric. It is not clear how heuristic choices influence the results qualitatively or quantitatively. Would e.g. running a PCA, and rescaling the first few components to represent environmental conditions yield the same results as plugging in all variables individually?

Analysis: Instead of analyzing the representativeness of each year by considering only flux tower sites operating at that year only I would find the cumulative effect, i.e. considering sites of this and previous years at least as relevant and informative. .

"Upscaling" The method is my opinion conceptually flawed. The authors consider a certain location, year, and month and then take the weighted mean of GPP of similar mean static environmental conditions for the same year and month. It could for example happen that for a pixel in the northern hemisphere in July flux measurements from the southern hemisphere with inverse seasonal cycle are considered because they appear similar in the space of static environmental conditions. Using static variables to map

seasonal and interannual variability is in itself highly questionable. The time varying coverage of FLUXNET data availability will create artefacts and discontinuities of the GPP product, which was also mentioned by the authors. The upscaling method itself is heuristic – there is no learning procedure involved to find a robust mapping for GPP based on environmental variables. There is no assessment of how the method works at flux tower sites. It would be super easy to do a leave one tower out cross-validation to evaluate how well it works – it's not shown and I guess it would show that it does not work. The authors' comparison against MTE-GPP for each year makes little sense in my opinion because spatial patterns of GPP won't change much between years (GPP will always be high in the tropics and low in dry and cold regions). If the authors wanted to compare interannual variability the mean per pixel should be removed to compare anomalies. Given that there are other approaches based on training machine learning algorithms on the market that make conceptually more sense, that have been evaluated, and are quite well established, I honestly do not see the point of proposing this method as a new useful approach for upscaling FLUXNET measurements.

Unjustified remarks and overstatements: There are many unjustified and overselling statements in the manuscript. For example: abstract (line 13): "optimal use"; page 10 (line 2-6; "Upscaling . . .[]"): sounds general but is not!; page 10 (line 26): "preserve variability" – I believe the method is not preserving variability but introducing a lot of artefacts. Page 13 (line 12-14): "our approach is better"; page 13 (line 30): again, there is no evidence that this method is preserving any useful variability; page 17 (line 4-6): "exhibits year to year variability in accuracy".

Minor points: Adding equations on top of describing the maths with text is necessary

Jung et al 2009 seems to be the wrong paper for the MTE product which is introduced in Jung et al 2011

---

## Referee Comment (RC4) · Anonymous Referee #4 · 6 Nov 2016

**Title: "Understanding the representativeness of FLUXNET for upscaling carbon flux from eddy covariance measurements"**

by Kumar et al.

**General Comments**

The main question addressed by the research is how well global terrestrial carbon cycling with the atmosphere can be represented over time when informed by time-resolved FLUXNET eddy-covariance observations and static delineations of eco-climatic regions. Corresponding goals are to understand the spatio-temporal representativeness of FLUXNET, to explore the patterning of inter-model agreement, and to identify eco-climatic regions that warrant future focus. The global mapping of FLUXNET carbon cycle observations is achieved through inverse distance weighted interpolation: The Euclidian distance between a target pixel and suitable FLUXNET sites is determined in multi-dimensional climate state-space, and not as a function as geographic proximity. One main difference to similar studies mapping the global carbon cycle is that no additional time-resolved drivers such as satellite data are used. The authors conclude that the study provides a window into the evolution of FLUXNET for representing the global carbon cycle, and that lower uncertainty is attained in the ecoregions that are well sampled.

The manuscript provides an incremental advance to the field of spatial representativeness and carbon cycle mapping. While the central questions are interesting and important, the manuscript essentially applies established methods to a new/different dataset. While the authors present conclusions about uncertainty metrics, these are not actually substantiated through quantification (details in the specific comments). To permit evaluating the significance and usefulness of the presented approach I strongly recommend a gridded uncertainty estimate from bootstrapping or propagation from cross-site validation. Special attention should be paid to the facts that (i) the Euclidean distance calculation does not appear to take into account inter-annual differences in environmental conditions, and (ii) FLUXNET sites are being switched on/off for interpolation based on monthly data availability. Hence, in the current form the results only partially contribute to and support the question and conclusions.

The manuscript clearly fits the aim of ESSD to further "the reuse of high-quality data of benefit to Earth system sciences". However, in its current form two areas might exceed the journal scope, namely "any interpretation of data is outside the scope", and "any comparison to other methods is beyond the scope". Hence, I recommend to eliminate corresponding areas from the manuscript, and use the freed space for a thorough quantification and discussion of uncertainty. The manuscript length is appropriate in general, although Figure 1 and Figure 6 are partially redundant and can be combined. Writing and language is fluent, and no copy-editing is needed. However, it would be much desirable to provide sufficient information in the figure legends for interpretation without having to scour the entire text.

On this basis my recommendation is publishable in general, subject to major revisions.

**Specific Comments**

Please see in-text comments and edits below.

[revised manuscript text omitted]

**2.3 Ecoregionalization**

FLUXNET classifies the sites based on vegetation type they represent using IGBP classification scheme (IGBP, 1990, 1992) among cropland, closed shrubland, deciduous broadleaf forest, evergreen broadleaf forest, evergreen needleleaf forest, grassland, mixed forest, open shrubland, savanna, wetlands and woody savanna. While the IGBP classifications are primarily based

5   on vegetation, a wider range of environmental variables (bioclimatic, edaphic and topographic) were employed in our study. The multivariate clustering technique has been widely used in ecology and Earth sciences for delineation of ecoregions that are relatively homogeneous with respect to selected environmental characteristics (Hargrove and Hoffman, 2005; Bernert et al., 1997; Lark, 1998; Hessburg et al., 2000). A massively parallel k-means clustering analysis tool developed by Kumar et al. (2011) was used to derive custom ecoregions using a selected set of environmental variables (Table 1).

10   Figure 2 shows the global land area classified into 10 distinct ecoregions at ~ 4 km resolution, when defined quantitatively by the k-means clustering algorithm, using 20 bioclimatic, edaphic and topographic characteristics (Table 1). Quantitatively defined ecoregions correspond well with the major global biomes and serves as a reference landcover classification for this study.

**2.4 Representativeness of flux sites**

15   Representativeness of a measurement site is the extent to which the measurements collected at any given location and time represent the conditions at any other location and time. Eddy covariance flux towers measure a range of meteorological and flux variables. While the measurements captured by the sensors are direct measures of a small region in the footprint of the

**Commented [R6]:** •This appears to be short form of Nappo et al. (1982). Might want to cite.
•Nappo, C. J., Caneill, J. Y., Furman, R. W., Gifford, F. A., Kaimal, J. C., Kramer, M. L., Lockhart, T. J., Pendergast, M. M., Pielke, R. A., Randerson, D., Shreffler, J. H., and Wyngaard, J. C.: The workshop on the representativeness of meteorological observations, June 1981, Boulder, Colorado, Bull. Am. Meteorol. Soc., 63, 761-764, 1982.

[Figure]

tower, understanding the representativeness of the measurements for a broader landscape is critical for upscaling of point mea-
surements to the larger area. There has been a number of attempts to assess the representativeness of flux tower networks.
Hargrove et al. (2003) analyzed the representativeness of the AmeriFlux network using an ecoregion based approach. For each
ecoregion in the map, they calculated the Euclidean distance in their data space to the single closest ecoregion that contains a
5  site from the network. Yang et al. (2008) studied the representativeness of the AmeriFlux network using MODIS and GOES
data and used Euclidean distance to quantify the similarity between any location and the collection of AmeriFlux sites. Using a
combination of footprint modeling and moderate-resolution remote sensing, Chen et al. (2011) calculated the representative-
ness with a sensor location bias metric. Sulkava et al. (2011) developed a tool to evaluate the representativeness of the European
eddy covariance network. They calculated representativeness as the Euclidean distance measured in data space between the
10  cluster centers and the established stations. He et al. (2015) studied the ChinaFlux sites and calculated the represenativeness
of flux tower sites as the Euclidean distance in a 4-dimensional environmental space. They calculated the Euclidean distance
from each pixel to each tower and selected the minimum distance as their network representativeness. Hoffman et al. (2013)
utilized climate and permafrost characteristics and derived a Euclidean-based representativeness for a sampling design in the
Arctic environment. They developed two approaches for calculating representativeness for sampling network design. First, the
15  ecoregions-based representativeness, where the dissimilarity of any ecoregion containing a sampling site to any other ecore-
gion, was calculated as the Euclidean distance between the two ecoregion centroids within the standardized n-dimensional data
space. The second was a point-based representativeness approach in which the Euclidean distance of each map pixel containing a
site from every other pixel on the map was calculated in the standardized n-dimensional data space. When a network of sites is
present, the Euclidean distance for the site closest to the pixel in data space was selected as the representativeness of that
20  pixel.

In this study, we used the point-based representativeness approach (Hoffman et al., 2013) to compute the dissimilarity of en-
vironmental conditions between any flux site location and every land pixel on the globe at ~4 km resolution in 20-dimensional
data space (Table 1). At each flux site the environmental characteristics (Table 1) were extracted from the gridded data sets
and the Euclidean distance to every land pixel on the globe was computed as $d\left(V^{site}, V^{pixel}\right) = \sqrt{\sum_{n=1}^{20}\left(V_n^{site} - V_n^{pixel}\right)^2}$,
25  resulting in a map showing dissimilarity in environmental conditions sampled by the flux site location to that of other locations
on the globe. For example, a representativeness map for the Willow Creek, U.S. site (US-WCr) (Figure 3) located in an upland
Deciduous Broadleaf Forest (DBF) shows that the site is representative of temperate environment in central United States and
Europe. However, this site does not capture well the dry environment of the southwestern U.S., coastal regions of the southern
U.S., evergreen forests in the northwestern U.S. Tropical and high latitude regions of the globe experience significantly differ-
30  ent environmental conditions and thus show high dissimilarities when compared to the US-WCr site. Similarly, the Evergreen
Broadleaf Forest (EBF) site at Santarum km83, Brazil (BR-Sa3) (Figure 3) represents tropical environments well, but shows
high dissimilarities with temperate and high latitude regions.

Environmental conditions at any location on the map may be characteristic of one or many or none of the sites in the sampling
network. To understand the best representative from the network as a whole, every pixel was assigned the representativeness
35  corresponding to the site closest to it in the multivariate environmental data space. Figure 4 shows the best representativeness of

[Figure]

**Commented [R7]:** •Which varies itself in space over time. Is this transient footprint bias taken into consideration here? If not an estimate of its impact on the upscaling exercise would be very helpful.

**Commented [R8]:** •It would help to present the approaches in order of the spatial scale they address, and explain their differing objectives on this basis.
•Please also see comment in introduction section.

**Commented [R9]:** •Here I get it: A proxy based on multi-dimensional space is used to determine the flux network representativeness for global coverage.
•I would help the reader to define this more clearly already in the introduction.
•The question that remains is: how close is close enough, i.e. what uncertainty is incurred for pixels that have a less close site than others?

[Figure]

[Figure]

[Figure]

(a) Willow Creek, U.S. (US-WCr)

[Figure]

(b) Santarum km83, Brazil (BR-Sa3)

Figure 3. Representativeness of individual flux sites.

**Commented [R10]:** •It would be good to provide sufficient information in the legend so the figure can be interpreted without having to scour the text for relevant information.

[Figure]

[Figure]

Figure 4. Network representativeness for all of the FLUXNET2015 sites (164 sites).

> **Commented [R11]:** •It would be good to provide sufficient information in the legend so the figure can be interpreted without having to scour the text for relevant information.

the FLUXNET network considering all 164 sites from FLUXNET2015. FLUXNET2015 shows fairly good representativeness for the large parts of the world except for northern high latitude regions like Iceland, wet tropical forests in South America and southeast Asia.

However, Figure 4 is only the theoretical representativeness of 164 sites in FLUXNET2015, since not all sites are in active

5 operation or have data available each year. As the sites come and goes out of operation, and depending on if data is available or not, our sampling (and thus upscaled view) of terrestrial carbon flux is variable through time. We calculated the representativeness of the network every month considering which sites were active and had data available. Figure 5 shows snapshots of representativeness of the network at different times during the period 1996–2014. Global carbon fluxes were poorly sampled in 1996, when only eight sites were available globally, but has steadily improved since then as sampling sites have been added.

10 Many eddy covariance measurement sites were added in mid-latitude regions of Europe and North America and Australia, where most of the flux sites are now clustered. Compared to 2011 (Figure 5(c)), 2014 (Figure 5(d)) shows a significant decline in the representiveness of the network globally due to data from a fewer number of sites being available. While many of the sites may be in active operation during recent years, the data from them most probably have not been made available, increasing uncertainties of global, upscaled estimates of carbon flux and highlighting the importance of making data available quickly.

[Figure]

[Figure]

[Figure]

(a) Representatives of FLUXNET2015 in 1996 (8 sites)

(b) Representatives of FLUXNET2015 in 2006 (76 sites)

(c) Representatives of FLUXNET2015 in 2011 (91 sites)

(d) Representatives of FLUXNET2015 in 2014 (61 sites)

Figure 5. Evolution of the spatial representativeness of FLUXNET2015 through time.

**Commented [R12]:** •It would be good to provide sufficient information in the legend so the figure can be interpreted without having to scour the text for relevant information.

[Figure]

**2.5 Upscaling point measurements**

Quantitative representativeness of network sites can be utilized to upscale flux estimates to larger spatial and temporal scales for input to or evaluation of process modeling or for estimating landscape-scale characteristics (Hoffman et al., 2013). Upscaling eddy covariance constrained estimates of gross primary production (GPP) from a global network of flux sites requires 1)
5   identifying all sites that sample environments similar to any location, and 2) developing a statistical model to estimate GPP at all global land regions using estimates from point measurements at flux sites.

**2.5.1 Inverse distance weighted interpolation**

Interpolation is a process of using measurements about a process at a limited number of point locations to make estimates about the process at other, unmeasured locations. Inverse distance weighting (IDW) is a deterministic, nonlinear interpolation
10   technique that uses a weighted average of the measurements from nearby sample points to estimate the magnitude of variables at non-sampled locations. IDW interpolation is based on Tobler's first law of geography (Tobler, 1970) which states "everything is related to everything else, but near things are more related than distant things". The weight of a measurement at a particular point is assigned in the averaging calculation depending upon the sampled point's distance to the non-sampled location. While traditional spatial interpolation approaches often employ the distance in geographic space between sampled points and the
15   un-sampled location, in this study the Euclidean distance in multivariate environmental data space (representativeness of the sampling location) was used to calculate the weights. Thus, samples from locations closer in terms of environmental conditions were assigned higher weights in interpolation while information about geographical proximity was never used in the process.

An important aspect of the inverse distance weighting interpolation is the neighborhood of influence that determines the sampling points that are used for interpolating the value at any un-sampled location. In this study we systematically defined the
20   neighborhood of influence (described in Section 2.5.2) and thus identified the flux measurements to include in the interpolation at each spatial location and at each monthly time step during the study period.

High temporal resolution measurements at eddy covariance sites capture the temporal dynamics and variability (seasonal and interannual), allowing us to examine the influences of phenology, drought, heat waves, El Niño, length of growing season, presence/absence of snow on canopy-scale fluxes, etc. (Baldocchi et al., 2001). Statistical models built using all the data,
25   while representing the long term trends and mean climatology well, also tend to lose the fine temporal dynamics captured by measurements. To preserve the rich temporal dynamics in the flux measurements, we built the interpolation at monthly time steps using the data only from that month. This approach, distinct from many past studies, helps capture the temporal variability in the upscaled data set; however, it is also prone to exhibit a bias if a localized phenomena is experienced by a flux site.

**2.5.2 Identifying flux sites in similar environments**

30   While global terrestrial ecosystems are highly diverse and heterogeneous, point measurements at flux sites sample the environment in which they are located. These measurements are representative of similar conditions at geographic locations elsewhere. The network of global flux sites which are managed and operated by independent regional institutions provide a sparse and

**Commented [R13]:** •It is my understanding that such approach does not take into account inter-annual differences in environmental conditions (the proxy used to determine Euclidean distance) across space.
•If this is the case, how can a defensible time-resolved surface flux product be created if the mapping/interpolation operator is constant in time?
•This poses the question, why does the study not also employ time-resolved satellite data products to fill this gap?
•Also, for pixels with state-space combinations that are observed by many flux sites, the resulting uncertainty should be smaller compared to those observed by few flux sites.
•Hence, I think an uncertainty budget is indispensable for determining the accuracy and usefulness of the presented approach.

c Author(s) 2016. CC-BY 3.0 License.

[Figure]

[Figure]

Figure 6. Spatial distribution of FLUXNET sites across global ecoregions.

> **Commented [R14]:** •This figure is redundant and can be combined with Figure 1.

non-uniform sampling of the environment. While some regions and environments may be well sampled, others may be undersampled. Thus it is important to identify the flux sites in similar environments where GPP estimates were available for upscaling. We overlaid the locations of sites in FLUXNET over the quantitatively defined ecoregions (Figure 2) and identified the sites in each ecoregion. Figure 6 shows the distribution of 164 FLUXNET sites in the FLUXNET2015 collection globally

5   across various ecoregions. To estimate the GPP at any location, the sites located within the same ecoregion would provide the most relevant and representative measurements. Ecoregions in the mid-latitudes of North America and Europe are well sampled by the global network of flux sites, while sites in many of the other ecoregions are few to none. Figure 4 shows the best possible representativeness while using all 164 sites in the FLUXNET2015 collection. The actual availability of flux site measurements is spatio-temporally variable.

10     Table 2 shows the number of sites within each ecoregion each year during the period 1991–2014 for which data were available in the FLUXNET2015 collection. We carefully processed and identified the sites at which GPP estimates were available within each ecoregion monthly for use in the upscaled product. However, there were some ecoregions and time periods that were completely unsampled by the available sites. In such cases we identified the ecoregion most similar in multivariate data space (Table 1) that was sampled by at least one or more flux sites and used those data for extrapolation. The

15   accuracy of the GPP estimates in such cases is expected to suffer due to data limitation.

> **Commented [R15]:** •...and requires quantification.

3   Results and discussions

We used monthly time series of GPP estimates from eddy covariance observations taken at flux towers from FLUXNET2015 to develop upscaled gridded GPP data sets at ~ 4 km resolution. We compared the upscaled GPP time series data set to

[Figure]

[Figure]

Table 2. Active flux sites present in each ecoregion during 1991–2014. Data from the period 1991–1999 was not used in the study. Some ecoregions had no flux sites available during the period 2000–2014.

| Year | Total | Ecoregions | | | | | | | | | |
|------|-------|--------|-------|--------|--------|--------|-------|--------|--------|-------|-------|
|      |       | 1      | 2     | 3      | 4      | 5      | 6     | 7      | 8      | 9     | 10    |
|      |       | 14.27% | 4.19% | 14.65% | 12.19% | 17.84% | 3.12% | 12.21% | 14.82% | 5.45% | 1.25% |
| 1991 | 1  | 1  | 0 | 0 | 0 | 0 | 0 | 0 | 0  | 0 | 0 |
| 1992 | 1  | 1  | 0 | 0 | 0 | 0 | 0 | 0 | 0  | 0 | 0 |
| 1993 | 1  | 1  | 0 | 0 | 0 | 0 | 0 | 0 | 0  | 0 | 0 |
| 1994 | 1  | 1  | 0 | 0 | 0 | 0 | 0 | 0 | 0  | 0 | 0 |
| 1995 | 2  | 2  | 0 | 0 | 0 | 0 | 0 | 0 | 0  | 0 | 0 |
| 1996 | 8  | 8  | 0 | 0 | 0 | 0 | 0 | 0 | 0  | 0 | 0 |
| 1997 | 10 | 9  | 1 | 0 | 0 | 0 | 0 | 0 | 0  | 0 | 0 |
| 1998 | 13 | 12 | 1 | 0 | 0 | 0 | 0 | 0 | 0  | 0 | 0 |
| 1999 | 15 | 14 | 1 | 0 | 0 | 0 | 0 | 0 | 0  | 0 | 0 |
| 2000 | 26 | 21 | 2 | 1 | 1 | 0 | 0 | 0 | 1  | 0 | 0 |
| 2001 | 40 | 26 | 2 | 3 | 1 | 0 | 0 | 0 | 3  | 0 | 5 |
| 2002 | 61 | 35 | 3 | 4 | 2 | 6 | 0 | 0 | 3  | 1 | 7 |
| 2003 | 73 | 37 | 3 | 7 | 5 | 8 | 1 | 0 | 3  | 1 | 8 |
| 2004 | 90 | 48 | 4 | 8 | 5 | 8 | 1 | 0 | 6  | 2 | 8 |
| 2005 | 89 | 49 | 3 | 8 | 5 | 5 | 1 | 1 | 7  | 2 | 8 |
| 2006 | 76 | 50 | 3 | 6 | 5 | 2 | 1 | 1 | 6  | 1 | 1 |
| 2007 | 80 | 51 | 5 | 5 | 5 | 3 | 1 | 1 | 8  | 1 | 0 |
| 2008 | 86 | 53 | 4 | 6 | 7 | 3 | 1 | 2 | 9  | 1 | 0 |
| 2009 | 89 | 54 | 4 | 4 | 6 | 3 | 1 | 2 | 14 | 1 | 0 |
| 2010 | 84 | 53 | 2 | 4 | 3 | 4 | 0 | 1 | 16 | 1 | 0 |
| 2011 | 91 | 57 | 3 | 3 | 3 | 2 | 0 | 3 | 18 | 2 | 0 |
| 2012 | 93 | 57 | 2 | 3 | 3 | 3 | 0 | 3 | 19 | 3 | 0 |
| 2013 | 85 | 51 | 2 | 3 | 3 | 2 | 0 | 3 | 18 | 3 | 0 |
| 2014 | 61 | 38 | 2 | 2 | 1 | 2 | 0 | 1 | 13 | 2 | 0 |

FLUXNET-MTE (Jung et al., 2009). While FLUXNET-MTE GPP was also developed using observations from FLUXNET sites, a different set of sites and data products were used in the current study. The overlapping period of 2000–2008 was used for comparison with FLUXNET-MTE.

3.1   Spatial distribution of GPP

5   Figure 7 shows the global spatial distribution of time integrated mean annual GPP for four years in the comparison period. The quality and accuracy of the upscaled GPP time series data is strongly affected by the availability and spatial distribution of the flux sites at any given time. Figure 8 shows the comparison of our upscaled GPP estimates with FLUXNET-MTE GPP estimates (Jung et al., 2009). Agreement was fairly good for Northern Hemisphere mid-latitudes regions that were well sampled

**Commented [R16]:** •Where is accuracy being quantified? It would be very helpful to show gridded uncertainty estimates.

[Figure]

by FLUXNET2015 sites. However, the data show significant differences in Southern Hemisphere tropical regions, which lack sufficient flux measurements. For example, FLUXNET-MTE employs the data for the flux sites in Matto Grosso (BR-Mtg) in South America, which was not available as part of FLUXNET2015 data set, leading to an underestimate of GPP in that region compared to FLUXNET-MTE. FLUXNET-MTE also suffers from limitations of data in this region, indicated by their high

5  index of extrapolation in tropical areas (Jung et al., 2009).

Extra-tropical regions in the Southern Hemisphere were poorly sampled by the FLUXNET2015 data with only a few sites making data available in that region. In addition, based on the environmental similarity sites in the Northern Hemisphere were used to estimate the fluxes in the region, which have an out-of-phase phenology, leading to an overestimation of GPP magnitude in Northern Hemisphere winter.

10  Higher estimates of GPP compared to FLUXNET-MTE in our upscaled data set in coastal regions of the Pacific Northwest of North America provide an improved representation of the wet evergreen forests. Coastal regions of the Pacific Northwest possess of high orographic relief and strong rain shadows. FLUXNET-MTE does not capture these highly productive forests, probably due to lower resolution averaging over wet and dry forests, but our approach at a high resolution provides a better estimate of GPP in this orographically complex terrain.

**Commented [R17]:** •Nice showcasing of how higher resolution can remedy non-linear aggregation effects.

15  ### 3.2 Seasonal patterns of GPP

Figure 9 shows the zonal mean seasonal patterns of monthly GPP estimated from FLUXNET2015 for years 2000, 2006, 2011, and 2014. In all cases, June and July exhibit the large GPP in the Northern Hemisphere extra-tropics. Weak seasonality, inter-annual variability, and regional under-sampling results in non-persistent ordering of monthly GPP in the tropics. As mentioned above, Southern Hemisphere high latitude regional estimates are strongly influenced by measurements in Northern Hemisphere

20  mid-latitudes because of the similarity in their environmental conditions and the lack of FLUXNET2015 measurement sites in the Southern Hemisphere. Consequently, anomalously high GPP is estimated during June and July in −30° to −50° latitude. The varying list of measurement sites each year also affects upscaled GPP estimates. For example, many fewer sites have made data from year 2014 available in FLUXNET2015, so Figure 9(c) exhibits unusually low tropical GPP. Figure 10(a) shows significant spread in our annual upscaled GPP because of both interannual variability and as a consequence of the continuously

25  evolving list of contributing measurement sites. However, it is likely that interannual variability is strong enough to result in great variability in GPP than is exhibited by the FLUXNET-MTE estimates in Figure 10(b).

**3.3 Temporal variability of GPP**

Eddy covariance observations at flux sites capture changes in biosphere–atmosphere exchange at a high temporal resolution. Temporal variability is due to long term changes in environmental conditions, vegetation dynamic, disturbance events, etc. To

30  preserve the temporal variability in GPP captured by the instruments in the upscaled product, only the observations from the same month were used in our IDW-based interpolation method.

Thus the pattern of time integrated annual GPP over the years show significant interannual variability for all latitudinal zones (Figure 10(a)). Variability was especially high in tropical region where only a few flux sites were present in FLUXNET2015

[Figure]

[Figure]

Figure 7. Spatial distribution of time integrated annual mean GPP fluxes [Units: $g\,C\,m^{-2}\,d^{-1}$].

[Figure]

[Figure]

Figure 8. Comparison of upscaled annual mean GPP fluxes to FLUXNET-MTE GPP. Positive values indicate higher estimates, and negative indicate lower estimated GPP compared to FLUXNET-MTE. [Units: g C m$^{-2}$ d$^{-1}$]

**Commented [R18]:** •Adding a relative scale would be helpful for straightforward interpretability.

[Figure]

[Figure]

Figure 9. Seasonal patterns of mean monthly GPP across latitudinal bands. [Units: $g\,C\,m^{-2}\,d^{-1}$]

data set. The magnitude and temporal variability of the upscaled GPP estimates were thus significantly influenced by regions where flux sites were present and may not be reflective of reality. Vegetation responses to the severe 2005 drought in the Amazon has been a topic of controversy with studies reporting patterns of greening (Saleska et al., 2007) and others reporting decline (Samanta et al., 2010). Our GPP product shows a green up during the 2005–2006 period in tropical zones; however,

5    this greening cannot be fully confirmed because of the lack of sufficient flux sites to adequately sample tropical regions for the spatio-temporal pattern of vegetation response to drought. For example, Santarem Km67 site (BR-Sa1) in Tapajos National Forest in Amazonia Brazil was one of the few sites in the tropics that was available in FLUXNET2015. Patterns of GPP observations at BR-Sa1 (Figure 11), including a greening trend during 2005-2006 drought, strongly influenced our upscaled data set. Flux measurements from more sites, including existing sites which are not available within FLUXNET2015, would

10    help improve the upscaled estimates of GPP.

[Figure]

[Figure]

| (a) Upscaled GPP | (b) FLUXNET-MTE |

Figure 10. Interannual variability in GPP fluxes for the period 2000–2008. [Units: $g\,C\,m^{-2}\,d^{-1}$]

[Figure]

Figure 11. Observed GPP timeseries at Santarem Km67 site (BR-Sa1), Brazil [Units: $g\,C\,m^{-2}\,d^{-1}$]

While FLUXNET-MTE shows a similar latitudinal pattern, interannual variability was significantly lower (Figure 10(b)). The FLUXNET-MTE model was trained using data across all years, and thus possibly was less susceptible to the strong interannual variability observed at the flux sites and rather is closer to a climatological mean pattern.

Due to variability in the number and spatial distribution of flux sites for which data were available in FLUXNET2015, 5   the upscaled GPP product also exhibits year to year variability in accuracy. Figure 12 and Table 3 summarize the statistical comparison of the upscaled GPP with FLUXNET-MTE GPP as reference benchmark.

**Commented [R19]:** •I am not sure, because the mapping operator in the present manuscript appears to be driven by proxies on a climatic timescale, i.e. without resolved inter-annual variability.
•What however does introduce inter-annual variability in the current study is that sites informing the mapping operation are being switched on/off depending on data availability in the given month.
•So it remains unclear whether the larger variability is a natural phenomenon, or an artifact of the mapping operation.
•I suggest that the susceptibility of the proposed mapping operation be studied and quantified in an uncertainty budget which will allow to substantiate the provided interpretation.

**Commented [R20]:** •Requires further explanation, cannot be interpreted with the little text provided here and in the legends.

[Figure]

Figure 12. Statistical comparison of upscaled time integrated annual mean GPP with FLUXNET-MTE GPP.

Table 3. Comparison statistics for upscaled GPP data set compared to FLUXNET-MTE

| Year | RMSE | Correlations (R) |
|------|------|-------------------|
| 2000 | 1.959 | 0.675 |
| 2001 | 1.557 | 0.717 |
| 2002 | 1.717 | 0.669 |
| 2003 | 1.710 | 0.739 |
| 2004 | 1.652 | 0.763 |
| 2005 | 2.425 | 0.594 |
| 2006 | 1.974 | 0.661 |
| 2007 | 1.484 | 0.730 |
| 2008 | 1.512 | 0.724 |

**4  Conclusions**

Networks of global flux sites capture the dynamic terrestrial biosphere–atmosphere gas exchange and provides us insights into terrestrial ecosystem processes. In this study we have presented a method to assess the representativeness of the global network of flux sites, specifically those included in the FLUXNET2015 data set. Due to independent operation of member flux sites in the FLUXNET network, a wider range of variation exists in duration of operation, measurements being conducted and availability of data sets for open research. Thus unlike previous studies, we conducted our assessment to quantify the representativeness of the network through time. The analysis provides a window into the evolution of the global eddy covariance measurements over time, quantify the contribution of any individual sites, identify the regions that are better represented by addition of sites over time, and ones that continue to be under-represented.

Upscaling of measured carbon fluxes is key for exploiting the full potential of rich FLUXNET2015 observations beyond the tower footprints to landscape scale understanding of ecosystem processes. We also developed a representativeness-based

upscaling approach to develop monthly time series high resolution global gridded estimates of GPP. Our upscaling method utilized the observations only from flux sites that sampled similar environmental conditions. Spatio-temporal variability and accuracy of the upscaled GPP data set strongly reflects the spatio-temporal variability in distribution of flux observations. Thus, the data set has higher accuracy and low uncertainty in the ecoregions that are well sampled by the available flux sites, and

5    lower accuracy and higher uncertainty in the ecoregions where available observations are sparse to none. This study was an attempt to understand the global carbon cycle (specifically GPP) through the lens of eddy covariance measurements across the globe. This work highlights the regions that are poorly understood by the existing (and available) networks, like Southern Hemisphere in general and carbon rich tropical forest ecosystems in particular.

This approach can easily be applied to flux data as they become available in order to routinely derive upscaled carbon fluxes.

10   While the current study was focused on GPP fluxes, the same method can be applied to other carbon, water and energy fluxes.

While a larger number of sites are part of the full global FLUXNET network, full potential of the network to understand the global carbon cycle can be realized only when data from all the sites are made available for research and analysis. Our future efforts will focus on integration of data sets from other flux and forest inventory networks, like the Global Ecosystem Monitoring (GEM) and RAINFOR networks, especially to improve our understanding of biodiversity and ecosystem function

15   in carbon rich tropical forests.

Data availability

All data sets from this study have been archived (Kumar et al., 2016) as a part of Next Generation Ecosystem Experiments (NGEE) Tropics Collection of Carbon Dioxide Information Analysis Center (CDIAC), Oak Ridge National Laboratory and are publicly available at http://dx.doi.org/10.15486/NGT/1279968. All gridded data are distributed in a CF Compliant NetCDF

20   format.

Author contributions.  JK, FMH, WWH designed the study and developed the method to quantify network representativeness. JK procesed the data, developed analysis method/tools and upscaled data sets. JK, FMH, WWH and NC analyzed the data. JK led the writing of manuscript with contributions from all the authors.

Acknowledgements. This work was supported by the Next Generation Ecosystem Experiments Tropics (NGEE–Tropics) and ORNL Terres-

25   trial Ecosystem Science Scientific Focus Area projects, sponsored by the U.S. Department of Energy, Office of Science, Office of Biological and Environmental Research. This manuscript has been authored by UT-Battelle, LLC under Contract No. DE-AC05-00OR22725 with the U.S. Department of Energy. The United States Government retains and the publisher, by accepting the article for publication, acknowledges that the United States Government retains a non-exclusive, paid-up, irrevocable, world-wide license to publish or reproduce the published form of this manuscript, or allow others to do so, for United States Government purposes. The Department of Energy will provide public

30   access to these results of federally sponsored research in accordance with the DOE Public Access Plan (http://energy.gov/downloads/doe-
* * *
**Commented [R23]:** •There is not a single quantitative accuracy metric to be found in the manuscript, so I don't think this is a valid conclusion.

**Commented [R24]:** •Please quantify, are we talking 1%, 10%, 100%?

**Commented [R25]:** •This would make for a great sentence in the introduction. There it could be combined with the explicit notion that flux measurements are only combined with a mapping operator based on climatological state-space representations to arrive at gridded, global data product. And that one main difference to other studies is that no additional time-resolved proxies such as satellite data are being used.

[Figure]

public-access-plan). This research used resources of the Oak Ridge Leadership Computing Facility at the Oak Ridge National Laboratory, which is supported by the Office of Science of the U.S. Department of Energy under Contract No. DE-AC05-00OR22725.

This work used eddy covariance data acquired and shared by the FLUXNET community, including these networks: AmeriFlux, AfriFlux, AsiaFlux, CarboAfrica, CarboEuropeIP, CarboItaly, CarboMont, ChinaFlux, Fluxnet-Canada, GreenGrass, ICOS, KoFlux, LBA, NECC,

5    OzFlux-TERN, TCOS-Siberia, and USCCC. The FLUXNET eddy covariance data processing and harmonization was carried out by the ICOS Ecosystem Thematic Center, AmeriFlux Management Project and Fluxdata project of FLUXNET, with the support of CDIAC, and the OzFlux, ChinaFlux and AsiaFlux offices.

[Figure]

**Appendix A: Upscaled monthly GPP time series**

Figure 13 shows the upscaled monthly time series of spatial distribution of global GPP fluxes for the year 2000.

[Figure]

(a) January 2000

(b) February 2000

(c) March 2000

(d) April 2000

(e) May 2000

(f) June 2000

[Figure]

[Figure]

Figure 13. Spatial distribution of upscaled global monthly GPP for year 2000.

[Figure]

---

## Author Comment (AC1) · 3 Feb 2017

D. Papale
darpap@unitus.it

Dear Authors

I'm one of the people involved in the FLUXNET2015 dataset preparation and for this reason I'm happy to see that the data are used. My comment is not related to the paper scientific content but to an important data policy requirement. As reported in the FLUXNET2015 data policy (http://fluxnet.fluxdata.org/data/data-policy/) when the data are used it is requested "that every publication specify each site used with the FLUXNET-ID, data-years used" and alos when available a reference describing the site. This is the way a PI can track their data use and for this reason important to follow. I would appreciate if you could add this table in the manuscript.

Dario Papale

We thank Dr. Papale for the comment. Failing to include a table of FLUXNET sites used in the analysis was oversight on our part. We have updated our analysis to use the latest November 3, 2016 release of the FLUXNET2015 and have included a list of all the sites that were used in the manuscript and have included the same table below.

We also reached out to all the site Principal Investigators (PI) to get their permission as per the Tier 2 data policy and have included site specific citations and acknowledgements as suggested by the site PIs. We have also added the FLUXNET2015 data provider team as author on the paper to give appropriate credit to the investigators for the data sets that were included in our analysis.

| Site ID | Site Name | Start Year | End Year | Latitude | Longitude |
|---|---|---|---|---|---|
| AR-SLu | San Luis | 2009 | 2011 | -33.4648 | -66.4598 |
| AR-Vir | Virasoro | 2009 | 2012 | -28.2395 | -56.1886 |
| AT-Neu | Neustift | 2002 | 2012 | 47.1167 | 11.3175 |
| AU-Ade | Adelaide River | 2007 | 2009 | -13.0769 | 131.1178 |
| AU-ASM | Alice Springs | 2010 | 2014 | -22.283 | 133.249 |
| AU-Cpr | Calperum | 2010 | 2014 | -34.0021 | 140.5891 |
| AU-Cum | Cumberland Plains | 2012 | 2014 | -33.6133 | 150.7225 |
| AU-DaP | Daly River Savanna | 2007 | 2013 | -14.0633 | 131.3181 |
| AU-DaS | Daly River Cleared | 2008 | 2014 | -14.1593 | 131.3881 |
| AU-Dry | Dry River | 2008 | 2014 | -15.2588 | 132.3706 |
| AU-Emr | Emerald, Queensland, Australia | 2011 | 2013 | -23.8587 | 148.4746 |
| AU-Fog | Fogg Dam | 2006 | 2008 | -12.5452 | 131.3072 |
| AU-Gin | Gingin | 2011 | 2014 | -31.3764 | 115.7138 |
| AU-GWW | Great Western Woodlands, Western Australia, Australia | 2013 | 2014 | -30.1913 | 120.6541 |
| AU-How | Howard Springs | 2001 | 2014 | -12.4943 | 131.1523 |
| AU-Lox | Loxton | 2008 | 2009 | -34.4704 | 140.6551 |
| AU-RDF | Red Dirt Melon Farm, Northern Territory | 2011 | 2013 | -14.5636 | 132.4776 |
| AU-Rig | Riggs Creek | 2011 | 2014 | -36.6499 | 145.5759 |
| AU-Rob | Robson Creek, Queensland, Australia | 2014 | 2014 | -17.1175 | 145.6301 |
| AU-Stp | Sturt Plains | 2008 | 2014 | -17.1507 | 133.3502 |
| AU-TTE | Ti Tree East | 2012 | 2014 | -22.287 | 133.64 |
| AU-Tum | Tumbarumba | 2001 | 2014 | -35.6566 | 148.1517 |
| AU-Wac | Wallaby Creek | 2005 | 2008 | -37.4259 | 145.1878 |
| AU-Whr | Whroo | 2011 | 2014 | -36.6732 | 145.0294 |

| AU-Wom | Wombat | 2010 | 2014 | -37.4222 | 144.0944 |
|---|---|---|---|---|---|
| AU-Ync | Jaxa | 2012 | 2014 | -34.9893 | 146.2907 |
| BE-Bra | Brasschaat | 1996 | 2014 | 51.3092 | 4.5206 |
| BE-Lon | Lonzee | 2004 | 2014 | 50.5516 | 4.7461 |
| BE-Vie | Vielsalm | 1996 | 2014 | 50.3051 | 5.9981 |
| BR-Sa1 | Santarem-Km67-Primary Forest | 2002 | 2011 | -2.8567 | -54.9589 |
| BR-Sa3 | Santarem-Km83-Logged Forest | 2000 | 2004 | -3.018 | -54.9714 |
| CA-Gro | Ontario - Groundhog River, Boreal Mixedwood Forest | 2003 | 2014 | 48.2167 | -82.1556 |
| CA-Man | Manitoba - Northern Old Black Spruce (former BOREAS Northern Study Area) | 1994 | 2008 | 55.8796 | -98.4808 |
| CA-NS1 | UCI-1850 burn site | 2001 | 2005 | 55.8792 | -98.4839 |
| CA-NS2 | UCI-1930 burn site | 2001 | 2005 | 55.9058 | -98.5247 |
| CA-NS3 | UCI-1964 burn site | 2001 | 2005 | 55.9117 | -98.3822 |
| CA-NS4 | UCI-1964 burn site wet | 2002 | 2005 | 55.9117 | -98.3822 |
| CA-NS5 | UCI-1981 burn site | 2001 | 2005 | 55.8631 | -98.485 |
| CA-NS6 | UCI-1989 burn site | 2001 | 2005 | 55.9167 | -98.9644 |
| CA-NS7 | UCI-1998 burn site | 2002 | 2005 | 56.6358 | -99.9483 |
| CA-Oas | Saskatchewan - Western Boreal, Mature Aspen | 1996 | 2010 | 53.6289 | -106.1978 |
| CA-Obs | Saskatchewan - Western Boreal, Mature Black Spruce | 1997 | 2010 | 53.9872 | -105.1178 |
| CA-Qfo | Quebec - Eastern Boreal, Mature Black Spruce | 2003 | 2010 | 49.6925 | -74.3421 |
| CA-SF1 | Saskatchewan - Western Boreal, forest burned in 1977 | 2003 | 2006 | 54.485 | -105.8176 |

| | | | | | |
|---|---|---|---|---|---|
| CA-SF2 | Saskatchewan - Western Boreal, forest burned in 1989 | 2001 | 2005 | 54.2539 | -105.8775 |
| CA-SF3 | Saskatchewan - Western Boreal, forest burned in 1998 | 2001 | 2006 | 54.0916 | -106.0053 |
| CA-TP1 | Ontario - Turkey Point 2002 Plantation White Pine | 2002 | 2014 | 42.6609 | -80.5595 |
| CA-TP2 | Ontario - Turkey Point 1989 Plantation White Pine | 2002 | 2007 | 42.7744 | -80.4588 |
| CA-TP3 | Ontario - Turkey Point 1974 Plantation White Pine | 2002 | 2014 | 42.7068 | -80.3483 |
| CA-TP4 | Ontario - Turkey Point 1939 Plantation White Pine | 2002 | 2014 | 42.7102 | -80.3574 |
| CA-TPD | Ontario - Turkey Point Mature Deciduous | 2012 | 2014 | 42.6353 | -80.5577 |
| CG-Tch | Tchizalamou | 2006 | 2009 | -4.2892 | 11.6564 |
| CH-Cha | Chamau | 2005 | 2014 | 47.2102 | 8.4104 |
| CH-Dav | Davos- Seehorn forest | 1997 | 2014 | 46.8153 | 9.8559 |
| CH-Fru | Früebüel | 2005 | 2014 | 47.1158 | 8.5378 |
| CH-Lae | Laegeren | 2004 | 2014 | 47.4781 | 8.365 |
| CH-Oe1 | Oensingen grassland | 2002 | 2008 | 47.2858 | 7.7319 |
| CH-Oe2 | Oensingen2 crop | 2004 | 2014 | 47.2863 | 7.7343 |
| CN-Cha | Changbaishan | 2003 | 2005 | 42.4025 | 128.0958 |
| CN-Cng | Changling | 2007 | 2010 | 44.5934 | 123.5092 |
| CN-Dan | Dangxiong | 2004 | 2005 | 30.4978 | 91.0664 |
| CN-Din | Dinghushan | 2003 | 2005 | 23.1733 | 112.5361 |
| CN-Du2 | Duolun_grassland (D01) | 2006 | 2008 | 42.0467 | 116.2836 |
| CN-Du3 | Duolun Degraded Meadow | 2009 | 2010 | 42.0551 | 116.2809 |
| CN-Ha2 | Haibei Shrubland | 2003 | 2005 | 37.6086 | 101.3269 |
| CN-HaM | Haibei Alpine Tibet site | 2002 | 2004 | 37.37 | 101.18 |
| CN-Qia | Qianyanzhou | 2003 | 2005 | 26.7414 | 115.0581 |

| CN-Sw2 | Siziwang Grazed (SZWG) | 2010 | 2012 | 41.7902 | 111.8971 |
|--------|------------------------|------|------|---------|----------|
| CZ-BK1 | Bily Kriz forest | 2004 | 2014 | 49.5021 | 18.5369 |
| CZ-BK2 | Bily Kriz grassland | 2004 | 2012 | 49.4944 | 18.5429 |
| CZ-wet | CZECHWET | 2006 | 2014 | 49.0247 | 14.7704 |
| DE-Akm | Anklam | 2009 | 2014 | 53.8662 | 13.6834 |
| DE-Geb | Gebesee | 2001 | 2014 | 51.1001 | 10.9143 |
| DE-Gri | Grillenburg | 2004 | 2014 | 50.9495 | 13.5125 |
| DE-Hai | Hainich | 2000 | 2012 | 51.0792 | 10.453 |
| DE-Kli | Klingenberg | 2004 | 2014 | 50.8929 | 13.5225 |
| DE-Lkb | Lackenberg | 2009 | 2013 | 49.0996 | 13.3047 |
| DE-Lnf | Leinefelde | 2002 | 2012 | 51.3282 | 10.3678 |
| DE-Obe | Oberbärenburg | 2008 | 2014 | 50.7836 | 13.7196 |
| DE-RuR | Rollesbroich | 2011 | 2014 | 50.6219 | 6.3041 |
| DE-RuS | Selhausen Juelich | 2011 | 2014 | 50.8659 | 6.4472 |
| DE-Seh | Selhausen | 2007 | 2010 | 50.8706 | 6.4497 |
| DE-SfN | Schechenfilz Nord | 2012 | 2014 | 47.8064 | 11.3275 |
| DE-Spw | Spreewald | 2010 | 2014 | 51.8923 | 14.0337 |
| DE-Tha | Tharandt | 1996 | 2014 | 50.9636 | 13.5669 |
| DE-Zrk | Zarnekow | 2013 | 2014 | 53.8759 | 12.889 |
| DK-Eng | Enghave | 2005 | 2008 | 55.6905 | 12.1918 |
| DK-Fou | Foulum | 2005 | 2005 | 56.4842 | 9.5872 |
| DK-NuF | Nuuk Fen | 2008 | 2014 | 64.1308 | -51.3861 |
| DK-Sor | Soroe | 1996 | 2014 | 55.4859 | 11.6446 |
| DK-ZaF | Zackenberg Fen | 2008 | 2011 | 74.4814 | -20.5545 |
| DK-ZaH | Zackenberg Heath | 2000 | 2014 | 74.4732 | -20.5503 |
| ES-Amo | Amoladeras | 2007 | 2012 | 36.8336 | -2.2523 |
| ES-LgS | Laguna Seca | 2007 | 2009 | 37.0979 | -2.9658 |

| ES-LJu | Llano de los Juanes | 2004 | 2013 | 36.9266 | -2.7521 |
|--------|---------------------|------|------|---------|---------|
| ES-Ln2 | Lanjaron-Salvage logging | 2009 | 2009 | 36.9695 | -3.4758 |
| FI-Hyy | Hyytiala | 1996 | 2014 | 61.8475 | 24.295 |
| FI-Jok | Jokioinen | 2000 | 2003 | 60.8986 | 23.5135 |
| FI-Let | Lettosuo | 2009 | 2012 | 60.6418 | 23.9597 |
| FI-Lom | Lompolojänkkä | 2007 | 2009 | 67.9972 | 24.2092 |
| FI-Sod | Sodankyla | 2001 | 2014 | 67.3619 | 26.6378 |
| FR-Fon | Fontainebleau-Barbeau | 2005 | 2014 | 48.4764 | 2.7801 |
| FR-Gri | Grignon | 2004 | 2014 | 48.8442 | 1.9519 |
| FR-LBr | Le Bray (after 6/28/1998) | 1996 | 2008 | 44.7171 | -0.7693 |
| FR-Pue | Puechabon | 2000 | 2014 | 43.7414 | 3.5958 |
| GF-Guy | Guyaflux (French Guiana) | 2004 | 2014 | 5.2788 | -52.9249 |
| GH-Ank | Ankasa | 2011 | 2014 | 5.2685 | -2.6942 |
| IT-BCi | Borgo Cioffi | 2004 | 2014 | 40.5238 | 14.9574 |
| IT-CA1 | Castel d'Asso 1 | 2011 | 2014 | 42.3804 | 12.0266 |
| IT-CA2 | Castel d'Asso 2 | 2011 | 2014 | 42.3772 | 12.026 |
| IT-CA3 | Castel d'Asso 3 | 2011 | 2014 | 42.38 | 12.0222 |
| IT-Col | Collelongo- Selva Piana | 1996 | 2014 | 41.8494 | 13.5881 |
| IT-Cp2 | Castelporziano 2 | 2012 | 2014 | 41.7043 | 12.3573 |
| IT-Cpz | Castelporziano | 1997 | 2009 | 41.7052 | 12.3761 |
| IT-Isp | Ispra ABC-IS | 2013 | 2014 | 45.8126 | 8.6336 |
| IT-La2 | Lavarone2 | 2000 | 2002 | 45.9542 | 11.2853 |
| IT-Lav | Lavarone | 2003 | 2014 | 45.9562 | 11.2813 |
| IT-MBo | Monte Bondone | 2003 | 2013 | 46.0147 | 11.0458 |
| IT-PT1 | Parco Ticino forest | 2002 | 2004 | 45.2009 | 9.061 |
| IT-Ren | Renon | 1998 | 2013 | 46.5869 | 11.4337 |
| IT-Ro1 | Roccarespampani 1 | 2000 | 2008 | 42.4081 | 11.93 |

| IT-Ro2 | Roccarespampani 2 | 2002 | 2012 | 42.3903 | 11.9209 |
|---|---|---|---|---|---|
| IT-SR2 | San Rossore 2 | 2013 | 2014 | 43.732 | 10.291 |
| IT-SRo | San Rossore | 1999 | 2012 | 43.7279 | 10.2844 |
| IT-Tor | Torgnon | 2008 | 2014 | 45.8444 | 7.5781 |
| JP-MBF | Moshiri Birch Forest Site | 2003 | 2005 | 44.3869 | 142.3186 |
| JP-SMF | Seto Mixed Forest Site | 2002 | 2006 | 35.2617 | 137.0788 |
| MY-PSO | Pasoh Forest Reserve (PSO) | 2003 | 2009 | 2.973 | 102.3062 |
| NL-Hor | Horstermeer | 2004 | 2011 | 52.2404 | 5.0713 |
| NL-Loo | Loobos | 1996 | 2014 | 52.1666 | 5.7436 |
| PA-SPn | Sardinilla Plantation | 2007 | 2009 | 9.3181 | -79.6346 |
| PA-SPs | Sardinilla-Pasture | 2007 | 2009 | 9.3138 | -79.6314 |
| RU-Che | Cherski | 2002 | 2005 | 68.613 | 161.3414 |
| RU-Cok | Chokurdakh | 2003 | 2014 | 70.8291 | 147.4943 |
| RU-Fyo | Fyodorovskoye | 1998 | 2014 | 56.4615 | 32.9221 |
| RU-Ha1 | Hakasia steppe | 2002 | 2004 | 54.7252 | 90.0022 |
| RU-Sam | Samoylov | 2002 | 2014 | 72.3733 | 126.4978 |
| RU-SkP | Yakutsk Spasskaya Pad larch | 2012 | 2014 | 62.255 | 129.168 |
| RU-Tks | Tiksi | 2010 | 2014 | 71.5943 | 128.8878 |
| RU-Vrk | Seida/Vorkuta | 2008 | 2008 | 67.0547 | 62.9405 |
| SD-Dem | Demokeya | 2005 | 2009 | 13.2829 | 30.4783 |
| SE-St1 | Stordalen grassland | 2012 | 2014 | 68.3541 | 19.0503 |
| SN-Dhr | Dahra | 2010 | 2013 | 15.4028 | -15.4322 |
| US-AR1 | ARM USDA UNL OSU Woodward Switchgrass 1 | 2009 | 2012 | 36.4267 | -99.42 |
| US-AR2 | ARM USDA UNL OSU Woodward Switchgrass 2 | 2009 | 2012 | 36.6358 | -99.5975 |
| US-ARb | ARM Southern Great Plains burn site- Lamont | 2005 | 2006 | 35.5497 | -98.0402 |

| US-ARc | ARM Southern Great Plains control site- Lamont | 2005 | 2006 | 35.5465 | -98.04 |
|---|---|---|---|---|---|
| US-ARM | ARM Southern Great Plains site- Lamont | 2003 | 2012 | 36.6058 | -97.4888 |
| US-Atq | Atqasuk | 2003 | 2008 | 70.4696 | -157.4089 |
| US-Blo | Blodgett Forest | 1997 | 2007 | 38.8953 | -120.6328 |
| US-Cop | Corral Pocket | 2001 | 2007 | 38.09 | -109.39 |
| US-CRT | Curtice Walter-Berger cropland | 2011 | 2013 | 41.6285 | -83.3471 |
| US-GBT | GLEES Brooklyn Tower | 1999 | 2006 | 41.3658 | -106.2397 |
| US-GLE | GLEES | 2004 | 2014 | 41.3665 | -106.2399 |
| US-Goo | Goodwin Creek | 2002 | 2006 | 34.2547 | -89.8735 |
| US-Ha1 | Harvard Forest EMS Tower (HFR1) | 1991 | 2012 | 42.5378 | -72.1715 |
| US-IB2 | Fermi National Accelerator Laboratory- Batavia (Prairie site) | 2004 | 2011 | 41.8406 | -88.241 |
| US-Ivo | Ivotuk | 2004 | 2007 | 68.4865 | -155.7503 |
| US-KS1 | Kennedy Space Center (slash pine) | 2002 | 2002 | 28.4583 | -80.6709 |
| US-KS2 | Kennedy Space Center (scrub oak) | 2003 | 2006 | 28.6086 | -80.6715 |
| US-Lin | Lindcove Orange Orchard | 2009 | 2010 | 36.3566 | -119.8423 |
| US-Los | Lost Creek | 2000 | 2014 | 46.0827 | -89.9792 |
| US-LWW | Little Washita Watershed | 1997 | 1998 | 34.9604 | -97.9789 |
| US-Me1 | Metolius - Eyerly burn | 2004 | 2005 | 44.5794 | -121.5 |
| US-Me2 | Metolius-intermediate aged ponderosa pine | 2002 | 2014 | 44.4523 | -121.5574 |
| US-Me3 | Metolius-second young aged pine | 2004 | 2009 | 44.3154 | -121.6078 |
| US-Me4 | Metolius-old aged ponderosa pine | 1996 | 2000 | 44.4992 | -121.6224 |
| US-Me5 | Metolius-first young aged pine | 2000 | 2002 | 44.4372 | -121.5668 |

| US-Me6 | Metolius Young Pine Burn | 2010 | 2014 | 44.3233 | -121.6078 |
|--------|--------------------------|------|------|---------|-----------|
| US-MMS | Morgan Monroe State Forest | 1999 | 2014 | 39.3232 | -86.4131 |
| US-Myb | Mayberry Wetland | 2010 | 2014 | 38.0498 | -121.7651 |
| US-Ne1 | Mead - irrigated continuous maize site | 2001 | 2013 | 41.1651 | -96.4766 |
| US-Ne2 | Mead - irrigated maize-soybean rotation site | 2001 | 2013 | 41.1649 | -96.4701 |
| US-Ne3 | Mead - rainfed maize-soybean rotation site | 2001 | 2013 | 41.1797 | -96.4397 |
| US-NR1 | Niwot Ridge Forest (LTER NWT1) | 1998 | 2014 | 40.0329 | -105.5464 |
| US-Oho | Oak Openings | 2004 | 2013 | 41.5545 | -83.8438 |
| US-ORv | Olentangy River Wetland Research Park | 2011 | 2011 | 40.0201 | -83.0183 |
| US-PFa | Park Falls/WLEF | 1995 | 2014 | 45.9459 | -90.2723 |
| US-Prr | Poker Flat Research Range Black Spruce Forest | 2010 | 2014 | 65.1237 | -147.4876 |
| US-SRC | Santa Rita Creosote | 2008 | 2014 | 31.9083 | -110.8395 |
| US-SRG | Santa Rita Grassland | 2008 | 2014 | 31.7894 | -110.8277 |
| US-SRM | Santa Rita Mesquite | 2004 | 2014 | 31.8214 | -110.8661 |
| US-Sta | Saratoga | 2005 | 2009 | 41.3966 | -106.8024 |
| US-Syv | Sylvania Wilderness Area | 2001 | 2014 | 46.242 | -89.3477 |
| US-Ton | Tonzi Ranch | 2001 | 2014 | 38.4316 | -120.966 |
| US-Tw1 | Twitchell Wetland West Pond | 2012 | 2014 | 38.1074 | -121.6469 |
| US-Tw2 | Twitchell Corn | 2012 | 2013 | 38.1047 | -121.6433 |
| US-Tw3 | Twitchell Alfalfa | 2013 | 2014 | 38.1159 | -121.6467 |
| US-Tw4 | Twitchell East End Wetland | 2013 | 2014 | 38.103 | -121.6414 |
| US-Twt | Twitchell Island | 2009 | 2014 | 38.1087 | -121.653 |
| US-Var | Vaira Ranch- Ione | 2000 | 2014 | 38.4133 | -120.9507 |
| US-WCr | Willow Creek | 1999 | 2014 | 45.8059 | -90.0799 |

| | | | | | |
|---|---|---|---|---|---|
| US-Whs | Walnut Gulch Lucky Hills Shrub | 2007 | 2014 | 31.7438 | -110.0522 |
| US-Wi0 | Young red pine (YRP) | 2002 | 2002 | 46.6188 | -91.0814 |
| US-Wi1 | Intermediate hardwood (IHW) | 2003 | 2003 | 46.7305 | -91.2329 |
| US-Wi2 | Intermediate red pine (IRP) | 2003 | 2003 | 46.6869 | -91.1528 |
| US-Wi3 | Mature hardwood (MHW) | 2002 | 2004 | 46.6347 | -91.0987 |
| US-Wi4 | Mature red pine (MRP) | 2002 | 2005 | 46.7393 | -91.1663 |
| US-Wi5 | Mixed young jack pine (MYJP) | 2004 | 2004 | 46.6531 | -91.0858 |
| US-Wi6 | Pine barrens #1 (PB1) | 2002 | 2003 | 46.6249 | -91.2982 |
| US-Wi7 | Red pine clearcut (RPCC) | 2005 | 2005 | 46.6491 | -91.0693 |
| US-Wi8 | Young hardwood clearcut (YHW) | 2002 | 2002 | 46.7223 | -91.2524 |
| US-Wi9 | Young Jack pine (YJP) | 2004 | 2005 | 46.6188 | -91.0814 |
| US-Wkg | Walnut Gulch Kendall Grasslands | 2004 | 2014 | 31.7365 | -109.9419 |
| US-WPT | Winous Point North Marsh | 2011 | 2013 | 41.4646 | -82.9962 |
| ZA-Kru | Skukuza | 2000 | 2013 | -25.0197 | 31.4969 |
| ZM-Mon | Mongu | 2000 | 2009 | -15.4378 | 23.2528 |